# Parsing Netlists of Integrated Circuits from Images via Graph Attention Network

**DOI:** 10.3390/s24010227

**Published:** 2023-12-30

**Authors:** Wenxing Hu, Xianke Zhan, Minglei Tong

**Affiliations:** College of Electronics and Information Engineering, Shanghai University of Electric Power, Shanghai 201306, China; moonstarwork@gmail.com (W.H.); cosydeal@163.com (X.Z.)

**Keywords:** object detection, deep learning, graph convolutional neural network, image processing algorithm, link prediction

## Abstract

A massive number of paper documents that include important information such as circuit schematics can be converted into digital documents by optical sensors like scanners or digital cameras. However, extracting the netlists of analog circuits from digital documents is an exceptionally challenging task. This process aids enterprises in digitizing paper-based circuit diagrams, enabling the reuse of analog circuit designs and the automatic generation of datasets required for intelligent design models in this domain. This paper introduces a bottom-up graph encoding model aimed at automatically parsing the circuit topology of analog integrated circuits from images. The model comprises an improved electronic component detection network based on the Swin Transformer, an algorithm for component port localization, and a graph encoding model. The objective of the detection network is to accurately identify component positions and types, followed by automatic dataset generation through port localization, and finally, utilizing the graph encoding model to predict potential connections between circuit components. To validate the model’s performance, we annotated an electronic component detection dataset and a circuit diagram dataset, comprising 1200 and 3552 training samples, respectively. Detailed experimentation results demonstrate the superiority of our proposed enhanced algorithm over comparative algorithms across custom and public datasets. Furthermore, our proposed port localization algorithm significantly accelerates the annotation speed of circuit diagram datasets.

## 1. Introduction

A massive number of paper documents that include important information such as circuit schematics can be converted into digital documents by optical sensors like scanners or digital cameras. Converting them into digital format for computer processing is a highly challenging task. This process facilitates the rapid provision of extensive training datasets for intelligent design processes and assists enterprises in swiftly reusing and transforming designs. The indispensable aspect of these processes revolves around the connections among electronic components.

The earliest related research dates back to the 1980s, when Okazaki et al. [1] proposed an image processing algorithm based on cyclical structures. This algorithm could identify each component on the entire circuit diagram and its topological relationships. Although this algorithm was developed for scenarios involving digital circuit design, this paper used generalized nodes as the foundation for analyzing the circuit connection relationships, leading to significant improvements in recognition accuracy and speed.

Subsequently, Cheng [2], from the University of Messina, introduced, in 1993, a symbol recognition system based on hierarchical neural networks for the first time. This system was applied to automate the processing of electrical engineering drawings, achieving faster and more accurate identification and classification of component symbols.

In 1997, a group of researchers from Lucent Technologies and the University of Nebraska, Lincoln proposed a more complex recognition system [3]. This system aimed at identifying various engineering drawings, enabling the recognition and interpretation of symbols and connecting lines. The system exhibited high versatility and is applicable to both digital and analog circuits. However, this research was based on conventional methods and did not achieve ideal recognition accuracy for analog circuitry.

Subsequently, with the development of deep learning, there have been studies integrating it into the recognition of hand-drawn electrical schematics [4,5,6]. The most recent papers can be traced back to 2015, when De et al. [7] proposed a circuit image segmentation algorithm. This method utilized morphological operations and histogram analysis to separate connecting lines and circuit components in circuit diagrams. These algorithms, benefitting from the integration of deep learning, witnessed significant improvements in recognition accuracy and precision.

Meanwhile, in recent years, there has been a continuous integration of deep learning into the field of circuit design, advancing the level of intelligent design. Many researchers have focused on leveraging deep learning to assist in circuit design, and their main research directions can be divided into the following three aspects.

The first direction involves evaluating the performance of circuit layout designs. Liu et al. [8] utilized computer vision techniques and trained a CNN (3D convolutional neural network) to achieve circuit performance predictions based on layout outcomes. However, their method is only applicable to amplifier circuits. Subsequent research extended this work [9]. They used graph neural networks to learn the circuit’s connectivity and combined it with image inputs for predictions. This multimodal approach enhanced the accuracy of performance estimation. Nonetheless, the study’s applicability remains limited and has certain constraints.

The second direction involves extracting rules from manual layouts. WellGAN [10] utilizes GANs (generative adversarial networks) to predict the placement of manually well-designed structures, aiming to achieve better performance metrics (area and wirelength) while ensuring compliance with design rules for various processes. GeniusRoute [11], on the other hand, employs a generative encoder model to learn the rules of manual analog routing. It predicts routing regions and feeds them into a known automated routing engine. Similar to the principles of WellGAN, GeniusRoute deploys a VAE (variational autoencoder) to learn the techniques employed by simulation engineers in layout and wiring, ultimately predicting high-performance layout scenarios.

The third direction involves extracting circuit layout constraints. Liu et al. [12] initially proposed a fast symmetric constraint detection method based on graph matching; however, this method has limited robustness and is only applicable to specific circuit structures. This work was further extended in work [13] using graph neural networks, which enable rapid retrieval of subgraphs with similar structures, thereby enhancing the algorithm’s generality. Subsequently, Kunal et al. introduced a graph convolutional neural network called GANA (graph convolutional network-based automated netlist annotation for analog circuits) [14] for identifying subcircuits. This framework effectively detects layout constraints such as symmetry and matching with a certain level of robustness.

It can be observed that machine-learning-based methods have significantly surpassed traditional approaches in terms of both speed and accuracy [15]. However, these studies on intelligent circuit design face challenges due to insufficient training datasets. Moreover, research progress in rapidly annotating the netlists of analog integrated circuits has been relatively slow in recent years. To address these issues, this paper proposes a bottom-up graph encoding model. This model comprises three sub-modules: an electronic component detection module, a port coordinate localization module, and a link prediction module. This model can swiftly extract the connectivity topology of electronic components from images, achieving rapid and automatic annotation of circuit datasets and digitization of paper-based schematics, as illustrated in Figure 1. The main contributions of this paper can be summarized as follows:We proposed the DC-PAFPN feature fusion network and developed an electronic component detection model based on the Swin Transformer architecture.We introduced the SGL-VGAE encoding model rooted in the graph attention network and presented the S-mish activation function, enhancing the representation of node features in embedded graph data. This achieved outstanding performance in training graph convolutional neural models.We annotated two customized datasets: an electronic component detection dataset and a dataset capturing electronic component connectivity relationships. Additionally, we autonomously devised a port localization algorithm, facilitating the automatic and rapid annotation of graph datasets.

To gauge the efficacy of SGL and the S-mish activation function, we have conducted comprehensive experiments on a selection of classic graph datasets. The results achieved have consistently surpassed baseline outcomes, underscoring the robustness and effectiveness of our proposed approaches.

## 2. Related Work

From Figure 1, it can be observed that the network topology extraction model constructed in this paper primarily consists of three components: object detection, a port localization algorithm, and link prediction. Therefore, in this chapter, we will focus on presenting the latest advancements in the fields of object detection and link prediction.

### 2.1. Electronic Component Detection

Currently, deep-learning-based object detection can be categorized into two types of networks: one-stage and two-stage object methods.

One-stage methods offer significant advantages in terms of speed and real-time applicability. They directly predict the position and class of objects in images. Prominent one-stage methods include the YOLO [16] series, SSD [17], and the RefineDet [18] algorithm.

On the other hand, two-stage methods prioritize accuracy. In this category, the representative methods are from the R-CNN [19] series, such as Fast R-CNN [20] and Faster R-CNN [21]. In recent years, transformer-based models have had a significant impact on two-stage object detection. Inspired by the introduced attention mechanisms, numerous high-performing object detection network studies [22,23,24,25] have emerged across various domains. These studies have motivated us to integrate object detection networks with the field of circuitry. Among them, the DETR [26] model and Swin-T [27] model are notable representatives. The model based on Swin-T utilizes shift windows and local attention mechanisms to achieve high recognition accuracy while effectively reducing computational requirements. Given the objectives of this study, the Swin-T model has been chosen as the backbone for constructing the electronic component detection network.

### 2.2. Link Prediction

Graph link prediction [28] has widespread applications in various fields such as social relationship recognition, protein structure prediction in biology, and traffic network forecasting. It deals with graph-type data, which encapsulate the topological structure and node attributes of a graph. Currently, the most popular techniques rely on GAEs (graph autoencoders) [29], which are extensions of AEs (Autoencoders) [30].

Typical GAEs consist of two main components: an encoder and a decoder. The encoder typically utilizes GNNs (graph neural networks) [31] to encode the structure and node features of the graph into latent embeddings represented as $Z\in R^{N\times d}$, where *N* represents the number of nodes in the graph, and *d* represents the dimensionality of node features. After obtaining high-quality node embeddings through the encoder, a decoder is used to reconstruct the original graph structure.

However, despite the effectiveness of graph autoencoder models in learning low-dimensional node embeddings, they still have certain limitations. First, as the number of encoder layers increases, GAEs face challenges of exponential time complexity and computational costs when expanding the neighborhood of nodes. Additionally, GAE models built using traditional GNNs may not have sufficient capacity to provide meaningful node representations for link prediction tasks [32].

In recent years, many studies have attempted to address the limitations of GAE models by utilizing subgraphs and different sampling strategies to enhance their generalization capabilities. Cluster GCN employs graph clustering algorithms to sample relevant nodes within subgraphs, thereby restricting the feature aggregation process to the subgraph and significantly reducing computational complexity. Kipf and colleagues introduced a VGAE (variational graph autoencoder) [29], which utilizes a parallel encoding process to learn the mean and variance of node latent vector representations, aiming to constrain the encoding results as closely as possible to a standard normal distribution. However, this parallel structure does not consider local information within the graph. Building upon this, Vaibhav et al. [33] introduced a regularization technique based on RWR (Random Walks with Restart) [34] to enhance model embedding representations. They also combined Random Walks with the SkipGram [35] strategy to capture implicit topological structures among graph nodes, maximizing the co-occurrence probability of similar nodes. Based on this, two regularization network frameworks were established, namely RWR-GAE and RWR-VGAE, as illustrated in Figure 2.

## 3. Proposed Method

The simple example of this work is illustrated in Figure 3.

The complete algorithm workflow can be summarized as follows:(1)Feed the digital image Figure 3a into the electronic component detection network to obtain the categories of electronic components and their relative positions in the image. The detection results are shown in Figure 3b.(2)Further process the detection results using a port localization algorithm to obtain the component ports and their position coordinates. The visualized results are shown in Figure 3c.(3)Annotate the graph data connection relationship matrix. Label the component ports as graph nodes and combine coordinates with encoded sub-image data into node feature matrices. An illustrative depiction of the annotated result for a single dataset sample is shown in Figure 3d.(4)Train the link prediction model. At this stage, the input to the network is the graph dataset annotated in step 3, and the encoder in the model output’s node feature embeddings is represented as Z. After decoding Z, predict the connection probability, calculate the loss against the labeled true connection relationship labels, and perform gradient descent on the encoder model.(5)Using the trained encoding model for link prediction, the network’s input consists of pairs of circuit nodes and their corresponding node feature matrices. The output is the connection probability between the two component ports. The network output results and visualization are shown in Figure 3e. The upper part of Figure 3e represents the network’s output results, including the existence probability and information about test node pairs. The lower part of Figure 3e visualizes the output results annotated on the image.

The following subsections will provide detailed explanations of the relevant improvements for the target detection network and the link prediction network.

### 3.1. DC-PAFPN

Similar to the current mainstream object detection network frameworks, our overall architecture is illustrated in Figure 4. The network employs the Swin Transformer as the feature extractor, taking in digital images containing electronic schematic diagrams. After feature extraction, these features undergo merging using DC-PAFPN, our enhanced fusion network utilizing deconvolution for upsampling, preserving more positional and edge information. Ultimately, the fused features are utilized for prediction, generating the position and confidence of the detected bounding boxes.

The quality of the network predictions is determined by features from different layers. Shallow-layer features primarily capture texture and color information, while deep-layer features contain more semantic information. Different recognition tasks have varying demands for different features. Traditional PAFPNs (path aggregation feature pyramidal networks) build upon FPNs (feature pyramid networks) [36] by adding two additional extension paths that fuse the shallowest features with lower-resolution higher-level features, effectively preventing the loss of useful information.

**Figure 4 sensors-24-00227-f004:**
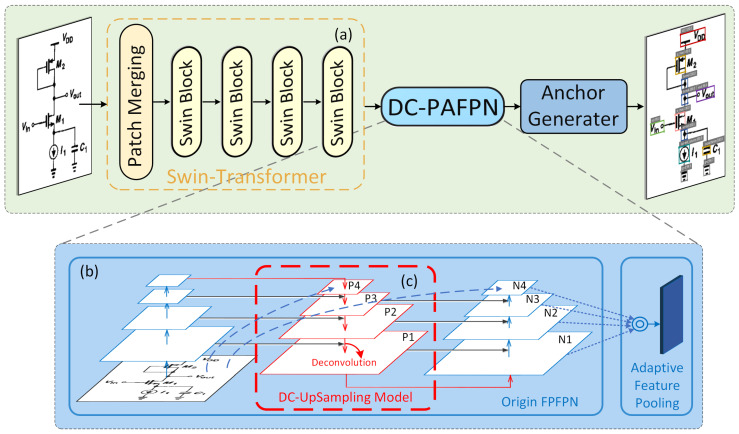
The architecture of electronic component detection network. (**a**) Swin-Transformer backbone. (**b**) The overall structure of PAFPN [37]. (**c**) Bottom-up path augmentation, which replaces the upsampling operation with deconvolution(transposed convolution) [38].

Figure 4b illustrates the improved PAFPN structure proposed in this paper, which shares a similar workflow with the traditional PAFPN. However, the main improvement lies in replacing the conventional nearest-neighbor interpolation in Figure 4c with a DC (deconvolution) upsampling module within the PAFPN framework. This adjustment aims to retain more positional and edge features. Here are the core principles of this method and its comparison with traditional upsampling algorithms:Adaptive Process of Deconvolution: The deconvolution process is trainable and allows for pixel-level classification. In simple terms, the parameters, specifically the convolutional kernel, actively adjust during gradient descent. They can automatically switch to different convolutional kernels to adapt to various scenarios. This adaptability enables the network, using deconvolution for upsampling, to learn more suitable coefficients based on data characteristics. This compensates, to some extent, for the reduction in receptive field caused by pooling and convolution operations, resulting in more accurate restoration of edge features and positional information within the image.Bilinear Interpolation: This method derives coefficients from pixel sampling positions and often leads to image edge blurriness.Nearest-Neighbor Interpolation: While computationally simple and fast, this method overlooks pixel continuity and variation during pixel filling, leading to distortions and artifacts.

This paper focuses on detecting electronic components within digital images highly sensitive to edge and positional information. Using traditional upsampling algorithms in this process might cause the loss of edge features, distortions, and artifacts, significantly affecting detection accuracy. The use of the deconvolution pathway effectively avoids the issues associated with these traditional methods. Figure 5 displays a comparison between feature embeddings obtained from the deconvolution pathway and traditional upsampling pathways.

In Figure 5, we randomly selected two sub-images to illustrate the differences between DC-PAFPN and traditional FPN. In this illustration, Figure 5a represents digital images fed into the target detection network, randomly selected from the training set. Subsequently, Figure 5b displays feature maps generated after processing through the Swin-T feature extraction network, presented in the form of heatmaps. Following that, Figure 5c exhibits the heatmaps corresponding to low-resolution features calculated during the FPN computation after downsampling. Next, Figure 5d demonstrates feature heatmaps obtained after upsampling using the proposed DC deconvolution pathway presented in this paper. Finally, Figure 5e showcases feature heatmaps generated using traditional bilinear interpolation. The comparison between Figure 5d and Figure 5e highlights that, owing to the learnable parameters in the deconvolution, the parts used for identification in the upsampled features are more prominent. For instance, the terms ‘kernel’ and ‘resistor’ in the upper image and ‘n_mos’ and ‘current_source’ in the lower image are more distinctly visible. Compared to the bilinear interpolation algorithm, the deconvolution pathway, when choosing suitable parameters, effectively alleviates the problem of overly blurry feature edges after upsampling. For our detection objectives, precise contours can offer more accurate identification results. Additionally, it mitigates the issue of excessive influence between adjacent pixels during bilinear interpolation. Such issues as inaccuracies in identifying the position due to neighboring pixels forming blocks of colors are avoided.

### 3.2. Link Prediction Network

In the link prediction part, most of our improvements are made to the encoding model, as depicted in the block diagram in Figure 6.

#### 3.2.1. SGL Structure

SGL (side-GAT-linear) is a bypass structure composed of a GAT (graph attention network) [39] and a linear layer, designed to enhance the encoding capabilities of the VGAE model. Its structure is illustrated in Figure 7.

The traditional graph encoding model’s node feature aggregation process can be perceived as a weighted summation of node features. The aggregation process’s weights can be updated through training. Consider a fully connected graph structure with n nodes. Using the traditional graph encoding network for node feature aggregation can be described by the following formula:(1)$$f\left(\hat{x_{i}}\right)=\sum_{i=1}^{n}\vec{h_{i}}*x_{i}$$
where $\vec{h_{i}}$ is a coefficient matrix that can be updated through learning. $x_{i}$ corresponds to the feature matrix of each node. The traditional VGAE encoding model employs two parallel branches to learn node feature embeddings. However, the subsequent inputs originate from the same GAT layer (they share the first graph convolutional neural network). While this encoding network can constrain the output feature representations into mean and variance results, it still heavily relies on neighboring nodes’ feature representations. If this shared first-layer weight pattern is abandoned, the graph neural network would degrade into two simple classification networks, losing the essence of encoding.

Therefore, this paper proposes the SGL (side-GAT-linear) bypass network structure to enhance the node feature aggregation process in the graph encoding model. This structure utilizes two independent GAT layers to learn node embedding representations and augments the obtained node features through two linear layers with updatable parameters. Finally, the output from the bypass structure is combined with the traditional VGAE model’s output, achieving reconstruction in the VGAE model. The convolutional process of the linear layer differs from graph convolution; it is not restricted by neighboring node features, assisting the network in better learning the graph’s latent topological structure and making more precise predictions.

The improved feature aggregation process can be simply described as:(2)$$f\left(\hat{x_{i}}\right)=\sum_{i=1}^{n}\vec{h_{i}}*\left(x_{i}+b_{i}\right)$$

The $b_{i}$ term stems from the expansiveness provided by the linear layer. In theory, this outcome allows for a portion of the node feature representation to exhibit any possible result, rather than continuing to be confined by neighboring node features. Thus, from a results standpoint, this bypass effectively expands both the upper and lower limits of graph node feature embedding representations, significantly enhancing the flexibility of graph feature extraction. It enables the network to handle more complex graph structures and adapt to various node connectivity scenarios.

#### 3.2.2. S-Mish Activation Function

The S-mish activation function in Figure 7 is a modification of the Mish function [40], where Tanh is replaced with the hyperbolic secant (sech) function. It is called S-mish. The formula for S-mish is as follows:(3)$$\hat{y}=x*\mathrm{sech}\left(Softplus\left(-ax\right)\right)$$

The expansions of sech(x) and Softplus(x) are given as follows:(4)$$\mathrm{sech}\left(x\right)=\frac{2}{exp\left(x\right)+exp\left(-x\right)}$$
(5)$$Softplus\left(x\right)=\log_{}\left(1+exp\left(x\right)\right)$$

The parameter *a* controls the shape of the function, regulating the inflow of negative gradients into the network. Specifically, as *a* increases, the amount of negative gradient inflow into the function decreases. Simultaneously, the function closely approximates the linear function y=x near the positive axis and introduces a certain amount of negative output near the negative axis. Importantly, as *a* tends toward infinity, the function transforms into the ReLU [41] activation function. This function maintains continuity across the entire real number domain. Figure 8 illustrates the curves of this function under different parameter values of *a*.

The function can be dissected through separate analyses of its positive and negative halves. For the scenario where x>0, the limit of the function expansion can be represented as follows:(6)$$f\left(x\right)=x*\mathrm{sech}\left(\log_{}\left(1+\exp\left(-ax\right)\right)\right)$$
(7)$$\lim_{x\to n\left(n>0\right)}f\left(x\right)=\frac{2n*\exp\left(2na\right)+2n*\exp\left(na\right)}{2*\exp\left(2na\right)+2*\exp\left(na\right)+1}$$

By performing a simple transformation on its form, we obtain:(8)$$\lim_{x\to n\left(n>0\right)}f\left(x\right)=n-\frac{n}{2*\exp\left(2na\right)+x*\exp\left(na\right)+1}$$

This expression elucidates why the function approximates the attributes of y=x for positive values of *x*. With the parameter *a* approaching infinity, the second term within the expression converges towards zero. Consequently, the curve along the positive half-axis deteriorates into the shape of the ReLU function.

Turning attention to the scenario where x<0, the limit of the function expansion can be described as follows:(9)$$\lim_{x\to n\left(n<0\right)}f\left(x\right)=\frac{2n*\exp\left(-na\right)+2n}{\exp\left(-2na\right)+2*\exp\left(-na\right)+2}$$

By performing a simple transformation on its form, we obtain:(10)$$f\left(n\right)=-n+\frac{n}{{\left(1+\exp\left(an\right)\right)}^{2}+\exp\left(an\right)}$$

This can be viewed as the sum of two functions:(11)$$y_{1}=-x$$
(12)$$y_{2}=\frac{x}{{\left(1+\exp\left(ax\right)\right)}^{2}+\exp\left(ax\right)}$$

The conclusion drawn is that the activation function devised in this paper has a limit of zero on the negative half-axis and is not contingent on the parameter *a*. Put simply, this activation function averts converting negative inputs into positive gradients, thus preventing gradient explosions. Moreover, graph-type data may encounter issues related to scale-free characteristics, where the degrees of nodes in a graph network might suffer from severe uneven distribution [42]. If only positive gradients flow in, it can lead to difficulties in convergence for some nodes’ outcomes. In essence, adjusting the parameter to control the inflow of negative gradients can enhance information continuity, which is highly effective for training graph models.

## 4. Two Custom Datasets

### 4.1. Electronic Component Detection Dataset

There is currently no publicly available dataset containing a large number of schematic diagrams of analog integrated circuits for object detection. Therefore, we have annotated a custom dataset specifically designed for electronic component detection. The content of this dataset is sourced from Razavi’s book *Analog CMOS Integrated Circuit Design* and includes various types of data, such as PDFs, scanned images, and photos. Figure 9 displays some annotated samples from the dataset.

The annotation process was carried out using the open-source dataset annotation tool Labelme [43]. We employed rectangular bounding-box annotations, which were later converted into the standardized COCO [44] format to simplify the training process. The dataset comprises a total of 1200 samples, covering 14 commonly used electronic component types, including NMOS, PMOS, current source, voltage source, capacitor, inductor, and various other components. Table 1 provides a detailed breakdown of the count for each specific category.

Among these, ‘kernel’ represents the circuit nodes we annotated, which can assist us in circuit analysis. Additionally, to mitigate the model’s challenge with small object detection, we applied data augmentation using the copy–paste [45] algorithm.

### 4.2. Electronic Component Connection Graph Dataset

The electronic component graph dataset consists of 3521 sample datasets, all of which are sourced from Razavi’s *Analog CMOS Integrated Circuit Design* book. Similar to typical graph datasets, our dataset includes nodes, connection matrices X, and node feature vectors.

In this dataset, the nodes correspond to the ports of electronic components, and the connection matrix contains the interconnections between these nodes. The node feature matrix is a 32-dimensional vector. The node feature vector can be succinctly represented in the following format:(13)$$tensor_{feature}=\left(x_{i},y_{i},tensor_{port},tensor_{piccode}\right)$$
where
(14)$$tensor_{port}=\left(x_{port1},y_{port1},x_{port2},y_{port2},x_{port3},y_{port3},x_{port4},y_{port4}\right)$$

The visual representation of the feature matrix composition is depicted in Figure 10. The detailed acquisition process of its three components is as follows:(1)$\left(x_{i},y_{i}\right)$ corresponds to the coordinates of the detection box obtained through object detection, specifically the coordinates of the top-left corner $\left(x_{1},y_{1}\right)$ and the bottom-right corner $\left(x_{2},y_{2}\right)$, forming a four-element feature matrix.(2)$tensor_{port}$ corresponds to the coordinates generated by the port localization algorithm. In the case of a four-port device, its corresponding feature matrix is an eight-element tensor representing the coordinates of the four ports. For instance, considering the schematic representation of ‘vin’ in the illustration, since it has only one port, the localization for the non-existent three ports is set to 0. Consequently, the resulting feature matrix is represented as $\left(0,0,0,0,0,0,x_{4},y_{4}\right)$.(3)$tensor_{piccode}$ corresponds to the encoding of the subgraph image. This outcome involves extracting the corresponding subgraph image based on the results of object detection. Following convolution processing, it is eventually reduced to a 1×20 tensor, constituting an essential part of the features.

The mentioned port localization algorithm can be summarized in four steps, and its overall block diagram is shown in Figure 11:(1)Fill the detection boxes of certain components with the color black to ensure the connectivity of the circuit backbone.(2)Use connected component detection to return the largest connected component, which represents the backbone of the schematic.(3)Apply an image thinning algorithm [46] to convert a certain connectivity region into a skeleton structure with a width of only one pixel.(4)Use the subgraph port localization algorithm to determine the port coordinates. Assuming the width and height of a certain subgraph image are *W* and *H*, respectively, fill the background coordinates (1,1) to (*W*-1, *H*-1) with pure white masks. With the help of the thinning algorithm, extract only one pixel wide contours from the subgraph, which represents the desired port coordinates. The process diagram of the subgraph port localization algorithm is shown in Figure 12.

Currently, the dataset assumes that electronic components have only four ports, and if a port is not detected, its coordinates are defined as 0. The output of the target detection network consists of ten classes, with nine electronic components (excluding ‘amplifiers’) having a maximum of four ports, including the ‘Kernel’ used for topology structure analysis. Amplifiers typically exhibit only three ports in the schematics (excluding power and ground lines), thus setting four ports covers most requirements. Moreover, the proposed network in this paper demonstrates exceptional scalability, requiring minor adjustments to accommodate different port numbers.

Using the Torch Geometric library [47], the nodes, connection matrix, and feature matrix are encapsulated into a graph data object, represented as “data”, and stored in the .pt file format. As components cannot be connected to themselves, the dataset does not set diagonal entries to 1 [48].

Table 2 presents a comparison of some classic datasets in the field of graph link prediction alongside the custom dataset. Cora is a graph dataset composed of machine learning papers and their related connections. Citeseer, also comprising paper relationships, encompasses 3312 papers from six domains. Lastly, PubMed is a graph dataset composed of relationships among 19,717 diabetes-related papers. These three datasets are highly representative in the domain of link prediction and could be compared to the “minist” dataset in the field of deep learning. Although they do not contain content related to circuits, as graph data, they share significant commonalities.

At the same time, it is evident that there are many differences between our dataset and these three classic datasets. The dataset proposed in this paper has a higher number of nodes, and the feature matrix for each node is relatively smaller. These two aspects significantly influence the model’s fitting speed and training duration. The proportion between edges and nodes also varies, but generally, a larger ratio of edges to nodes tends to yield more precise prediction results. This is easily comprehensible; as the edge proportion increases, each node shares more features, leading to a more accurate model fit.

## 5. Experiment

### 5.1. Electronic Component Detection

#### 5.1.1. Evaluation Criteria

In the realm of object detection, the evaluation metrics encompass two principal aspects: mAP (mean average precision) and IoU (Intersection over Union). The mAP quantifies the average of AP (average precision) scores across multiple distinct classes. AP, in turn, denotes the average precision and is computed as the area under the precision–recall curve. An optimal mAP value stands at 1, and an elevated value signifies superior detection performance. Precision and recall are pivotal metrics gauging accuracy and recall, respectively. The procedure for their computation unfolds as follows:(15)$$\mathrm{Accuracy}=\frac{\mathrm{TP}+\mathrm{TN}}{N},$$
where N = TP + TN + FP + FN
(16)$$\mathrm{Recall}=\frac{\mathrm{TP}}{\mathrm{TP}+\mathrm{FN}}$$

The comprehension of these evaluation metrics necessitates a grasp of key terms:TP (true positive) represents the instances that are predicted as positive and are genuinely positive.TN (true negative) signifies the instances that are predicted as negative and are genuinely negative.FP (false positive) denotes the cases that are predicted as positive but are, in actuality, negative.FN (false negative) accounts for the instances that are predicted as negative but are truly positive.

Another metric for object detection is achieved through the use of Intersection over Union (IoU), which is calculated by dividing the intersection area of the predicted bounding box and the ground-truth bounding box by their union area. The formula for its computation is articulated as follows:(17)$$\mathrm{IoU}=\frac{\mathrm{Area}\left(C\right)\cap \mathrm{Area}\left(G\right)}{\mathrm{Area}\left(C\right)\cup \mathrm{Area}\left(G\right)}$$

The ideal scenario is when the intersection of the two bounding boxes is equal to their union, indicating that the predicted result perfectly matches the annotated ground truth.

#### 5.1.2. Analysis of Results

During network training, the size of the input image samples is resized to 1333 × 800. The dataset is split into a training set and a test set with a ratio of 7:3. Optimization is performed using Stochastic Gradient Descent (SGD) with dynamic learning rate scheduling. The initial learning rate is set to 0.01, and the batch size is set to 2. Over the course of 32 training iterations, the learning rate is reduced by a factor of 0.1 starting at the 8th and 11th iterations.

The evaluation of results hinges on several metrics, including mAP, $AP_{50}$, and $AP_{75}$, in addition to AP values for targets of varying sizes (large, medium, and small). mAP embodies the average precision for all classes, $AP_{50}$ stands for the AP concerning detection boxes with an IoU greater than 0.5, and $AP_{75}$ corresponds to the AP associated with detection boxes possessing an IoU surpassing 0.75. Notably, in comparison to the baseline, the proposed algorithm demonstrates improvements of 4.3 percentage points in mAP, 0.9 percentage points in $AP_{50}$, 4.8 percentage points in $AP_{75}$, and 4.5 percentage points in AP for small targets. Furthermore, an average enhancement of 0.8 percentage points is observed across other classes. The comprehensive results are meticulously documented in Table 3.

Throughout the experiments, each network underwent five sets of trials, and we averaged the results, presenting them in the table.

We reproduced the network models mentioned in Table 3 and conducted experiments on our custom dataset. The first two networks are representative one-stage models, where YoloX-Tiny [50] represents the latest research in the YOLO series, known for its high inference speed and respectable recognition accuracy. DDOD [51] is a dense object detection network evolved from SSD. The latter two networks are representative two-stage models. Sparse RCNNRCNN [52] represents the latest developments based on the RCNN algorithm, while Swin-T [27] serves as our baseline, both renowned for their exceptional detection accuracy.

The visual results of different models are illustrated in the following Figure 13.

Figure 13a,d,g are the results obtained from the network built by the original Swin and FPN. Figure 13b,e,h are the results obtained from DDOD. Figure 13c,f,i are the results obtained from the network built by Swin and the improved DC-PAFPN.

Among these results, the top three performing networks correspond to Table 3. Among them, DDOD’s performance stands out, even surpassing the original Swin-T. In fact, DDOD is a one-stage dense object detector inspired by SSD. It introduces deformable convolutions to improve spatial feature decoupling. Additionally, it separates the label assignment process for classification and regression tasks, introducing a hyperparameter α to balance the contributions of different labels to network training. This method better selects features conducive to network convergence. Visualizations indicate DDOD’s balanced detection accuracy, avoiding significant discrepancies in inter-class accuracy seen in combinations like Swin-T+FPN. Furthermore, DDOD exhibits excellent position localization, sometimes even surpassing the accuracy of the Swin-T+DC-PAFPN combination with improved feature fusion. However, being fundamentally a one-stage network, DDOD might lack in accuracy compared to the networks proposed in this paper. We not only merge contour information from shallow features with deep semantic information but also use deconvolution layers to provide precise positional information for the feature fusion process. Consequently, the obtained object detection results exhibit both accurate localization and exceptionally high recognition accuracy.

### 5.2. Link Prediciton

#### 5.2.1. Evaluation Criteria

The evaluation criteria for link prediction algorithms mainly use two indicators: AUC (area under the receiver operating characteristic curve) and AP (precision at N). For a trained connected prediction model, when inputting the feature matrix of two nodes, the possibility of an edge between the two nodes is output. At this point, AUC can be understood as a higher probability of predicting the actual existence of an edge than a randomly selected non-existent edge. The calculation process is based on comparing the network output values of existing and non-existent edges in the test set. The specific calculation process is as follows:If $test_{exit}>test_{absent}$, the numerator is incremented by 1, indicating a correct prediction.If $test_{exit}=test_{absent}$, the numerator is augmented by 0.5, signifying a random selection made by the model.If $test_{exit}<test_{absent}$, the numerator remains unaltered, indicating a random selection made by the model.

Extend the testing process described above to N times. If situation 1 occurs, record 1 point; if situation 2 occurs, record 0.5 points; if situation 3 occurs, record 0 points. The corresponding formula for calculating AUC is as follows:(18)$$\mathrm{AUC}=\frac{n^{\prime }+0.5n^{\prime \prime }}{N}$$

Another evaluation metric is AP (average precision), which focuses on the accuracy of the top *N* prediction results. After obtaining the likelihood of edges between nodes in the test set using the link prediction network, they are sorted based on likelihood. The top *N* results are selected, and if among those N results, K of them are actual edges in the validation set, the accuracy AP is calculated as K/N.

The two metrics focus on different aspects of measuring prediction accuracy. AUC provides an overall assessment of the algorithm’s accuracy, while AP specifically considers whether the top N edges are predicted accurately. Typically, link prediction primarily relies on the AUC result. In cases where AUC results are similar, AP can be used to further assess the algorithm’s accuracy.

#### 5.2.2. Analysis of Results

The experiments for link prediction were evaluated using the AUC and AP metrics. Additionally, extensive experiments were conducted on public datasets to demonstrate the robustness of the proposed model. By comparing and analyzing the results against the baseline, it can be concluded that the algorithm presented in this paper achieved an accuracy of AUC = 93.4 on the custom dataset, which is 1.2 percentage points better than the baseline. Furthermore, during the testing process on public datasets, the algorithm also achieved an average performance improvement of more than 1.7 percentage points over the baseline. For specific results, please refer to Table 4.

To ensure the accuracy of the experiments, each algorithm was repeated ten times, and the averages were taken as the final results. Additionally, Table 4 reflects the fluctuation in results for different algorithms. DeepWalk [53] is a network representation approach which encodes social relations into a continuous vector space by learning structural regularities present within short random walks. GAE∣VGAE is an effective approach for learning social embedding.This method generates a representation in $R_{d}$ from the d-smallest eigenvectors of *L*, the normalized graph Laplacian of G. ARGE∣ARVGE [54] is an adversarially regularized autoencoder algorithm which uses a (variational) graph autoencoder.

To further demonstrate the validity of the model’s results, a quantitative analysis of the network’s output was conducted using the following procedure:(1)Using the trained link prediction network to make predictions, where the input consists of pairs of nodes, and the output is the probability of an edge existing between the nodes.(2)Searching for the corresponding labels in the test set to determine whether there is a true connection between the pairs of nodes and matching them with the predictions.(3)Randomly sampling 64 data points from all the matched results and plotting them as scatter plots for comparison. The actual network prediction results comparison is shown in Figure 14.

Figure 14a corresponds to the scatter plot obtained by using the link prediction algorithm SLG-RWR-VGAE proposed in this paper to predict the custom dataset. Figure 14b represents the results of the baseline algorithm. It can be observed that the model proposed in this paper is more stable during the training process, and the results of predicting node connection probabilities are smoother, without significant fluctuations. Moreover, from the scatter plot, it is evident that the SLG-RWR-VGAE algorithm provides results much closer to 1 (the true value) when real connections exist, making it more accurate compared to the baseline algorithm, consistent with the earlier AUC and AP experimental results.

### 5.3. Ablation Experiments

To ascertain the efficacy of the enhancements introduced to the link prediction network, namely the GAT layer, the SGL (side-GAT-linear) bypass structure, and the S-mish activation function, ablation experiments were meticulously executed. The resultant findings are meticulously outlined in Table 5.

The results of the ablation experiments show that replacing the original GCN layer with the graph attention network (GAT) for encoding the input graph data significantly improved the effectiveness of link prediction. Additionally, the proposed SGL bypass structure in this paper greatly stabilizes the training process of graph models, ensuring consistent results across multiple training runs. It also helps the network mitigate the problem of feature embeddings overly depending on neighboring nodes, resulting in more effective node embeddings. Lastly, the S-mish activation function introduced in this paper allows some negative gradients to flow in, aiding the graph neural network in learning to retain more useful feature information. With the appropriate parameter settings, it can effectively enhance the accuracy of network link prediction.

## 6. Conclusions

The model proposed and tested in this research has the potential to rapidly annotate the topological structure of analog integrated circuits from image data, and it demonstrates high robustness. Its applications in the field of intelligent design for analog integrated circuits can significantly accelerate the design process and simplify the design of analog integrated circuits. Future research may focus on improving the accuracy of detecting and predicting connection relationships. Additionally, expanding the dataset to include different types of circuits can greatly enhance the model’s robustness.

Furthermore, the current research analysis is limited to the rapid annotation of connection relationships. Future work may delve into extracting electronic component parameters, achieving more comprehensive automation of netlist extraction. This avenue could be further explored through the prediction of component characteristics using node features in the field of graph modeling.

## Figures and Tables

**Figure 1 sensors-24-00227-f001:**
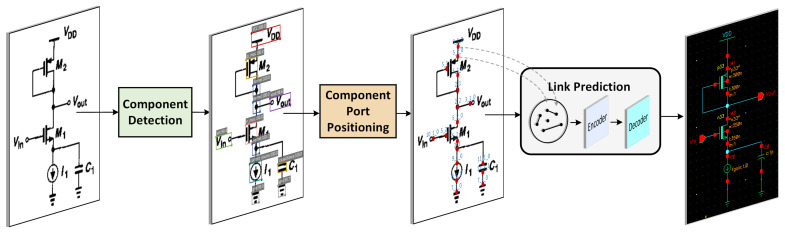
The block diagram of the overall process.

**Figure 2 sensors-24-00227-f002:**
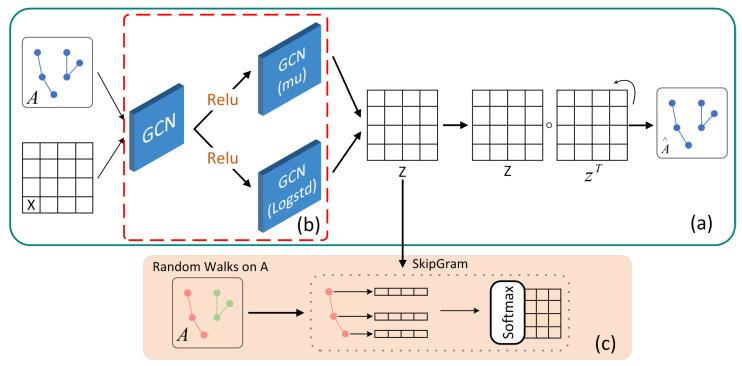
The diagram of RWR-VGAE. (**a**) Origin RWR-VGAE model, (**b**) origin VGAE, and (**c**) Random Walk regularized.

**Figure 3 sensors-24-00227-f003:**
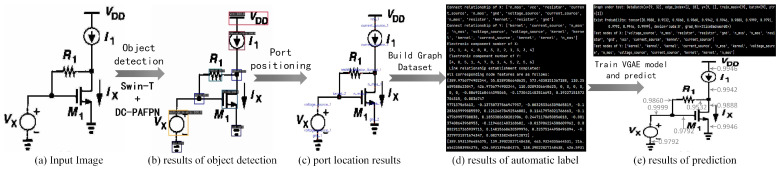
Simple example of our work.

**Figure 5 sensors-24-00227-f005:**
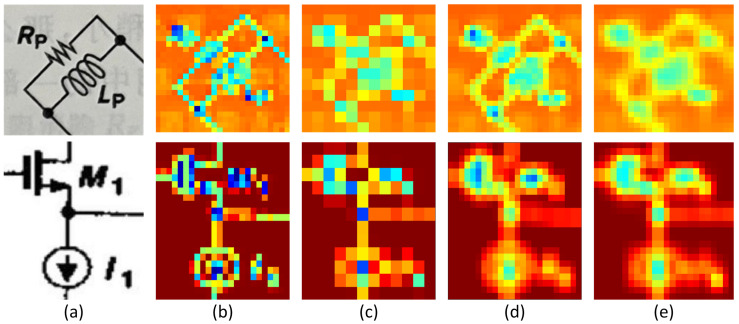
The architecture of electronic component detection network. (**a**) Input image. (**b**) Schematic diagram of extracted features. (**c**) Low-level image features. (**d**) Upsampling result obtained by deconvolution. (**e**) Upsampling result obtained by linear interpolation.

**Figure 6 sensors-24-00227-f006:**
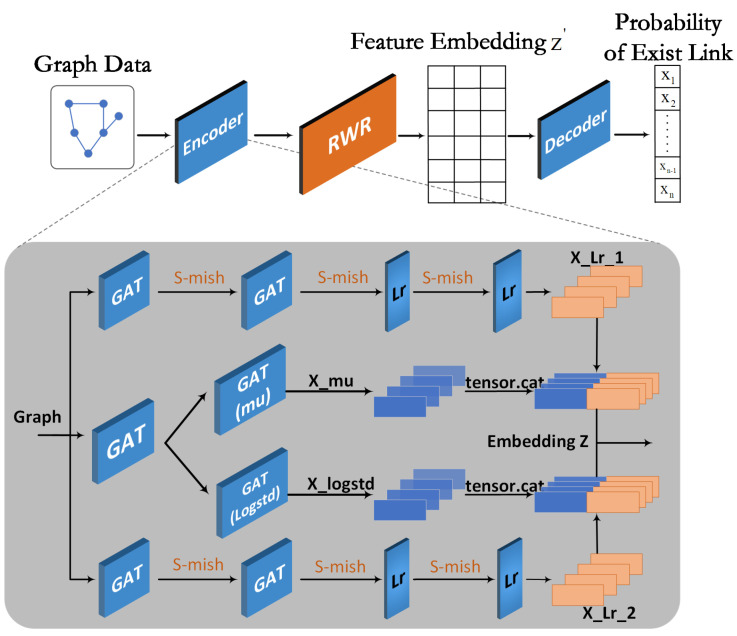
The schematic diagram of the Link prediction network.

**Figure 7 sensors-24-00227-f007:**
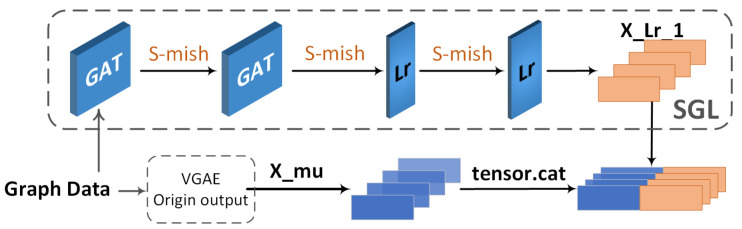
The schematic diagram of a network frame.

**Figure 8 sensors-24-00227-f008:**
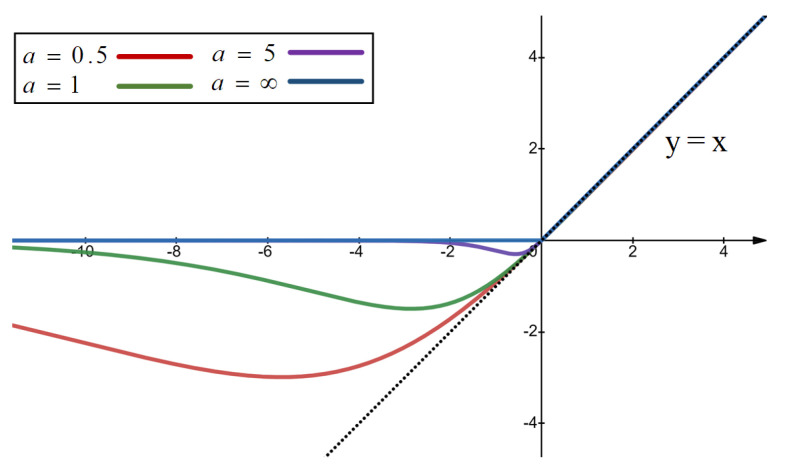
Graphs of the function with different parameters.

**Figure 9 sensors-24-00227-f009:**
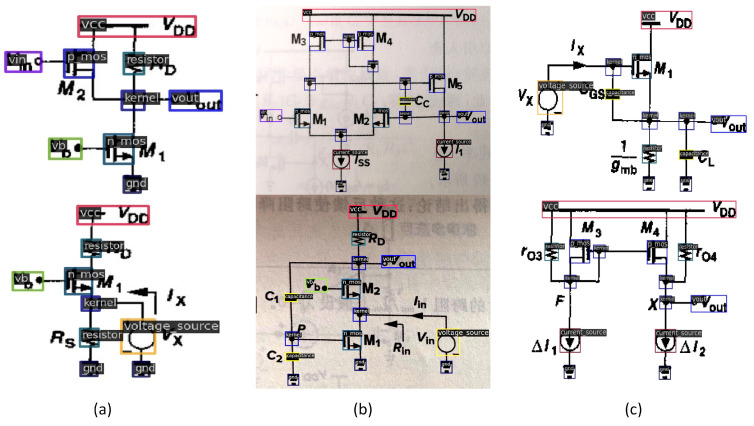
Schematic diagram of component testing and labeling samples. (**a**) PDF images. (**b**) Captured images. (**c**) Scanned images.

**Figure 10 sensors-24-00227-f010:**
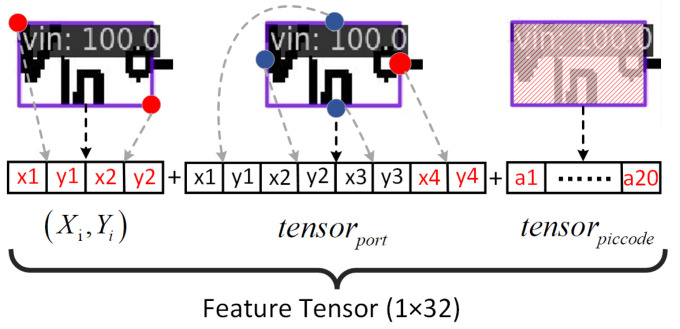
Schematic diagram of the construction of the feature tensor.

**Figure 11 sensors-24-00227-f011:**
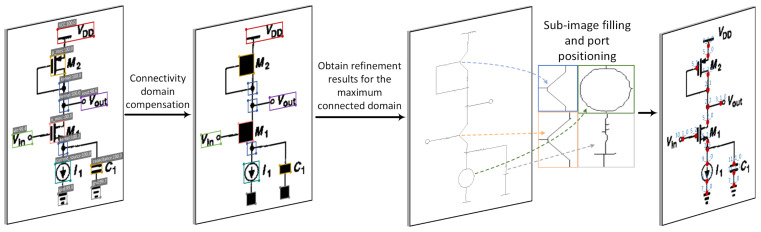
Port location algorithm flowchart.

**Figure 12 sensors-24-00227-f012:**
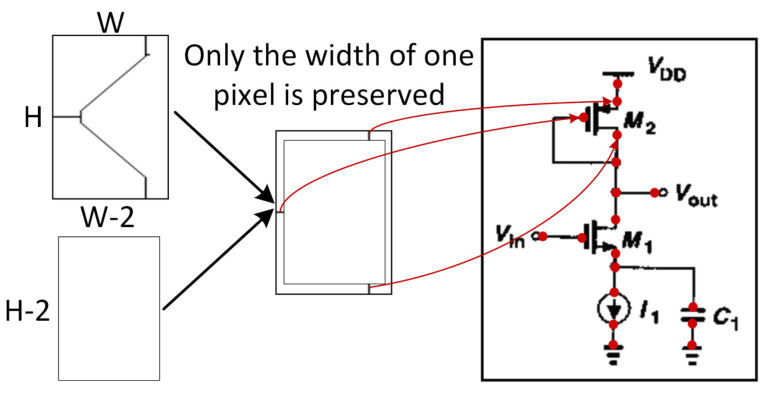
The diagram of background color filling process.

**Figure 13 sensors-24-00227-f013:**
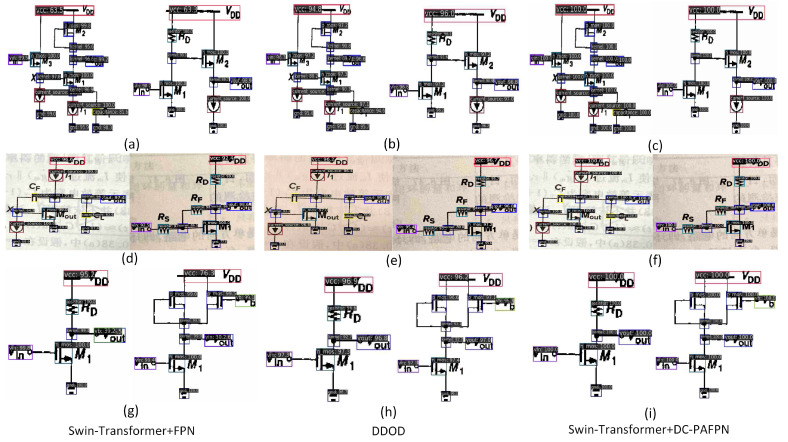
Qualitative comparison of object detection on custom dataset. (**a**) The result of Swin-T detecting PDF images. (**b**) The result of DDOD detecting PDF images. (**c**) Improved network detection of PDF images. (**d**) The result of Swin-T detecting captured images. (**e**) The result of DDOD detecting captured images. (**f**) Improved network detection of captured images. (**g**) The result of Swin-T detecting scanned images. (**h**) The result of DDOD detecting scanned images. (**i**) Results of improved network detection of scanned images.

**Figure 14 sensors-24-00227-f014:**
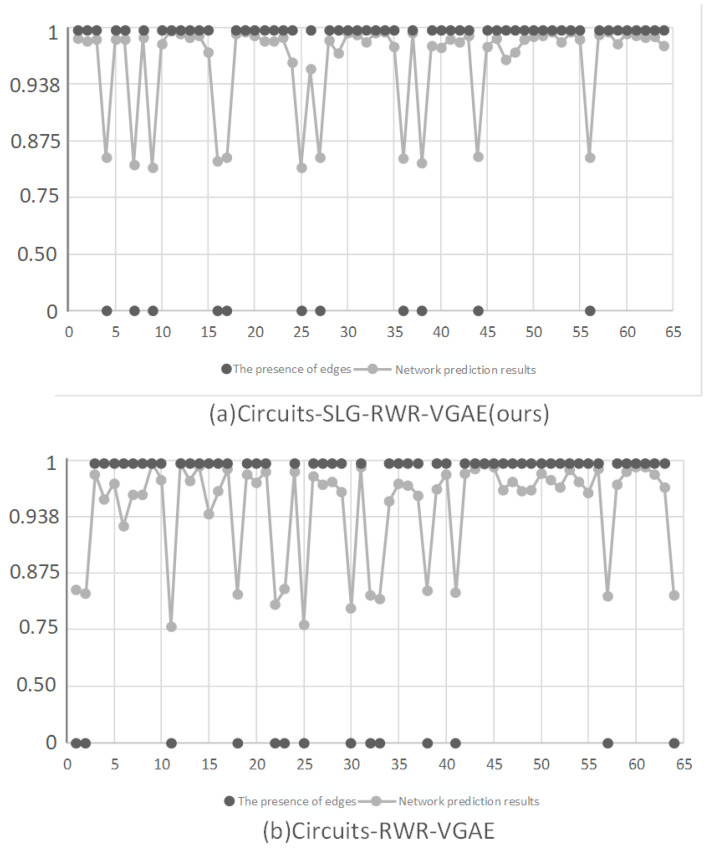
Scatter plot comparing actual network prediction results.

**Table 1 sensors-24-00227-t001:** Component testing dataset for single category number statistics.

vcc	current_source	vb	capacitance	kernel	vout	inductor
4292	3553	1922	3929	16,316	3471	2370
vin	power_amplifier	gnd	voltage_source	p_mos	n_mos	reistor
2743	820	2543	1027	3902	8755	5366

**Table 2 sensors-24-00227-t002:** Comparison table of linked forecast data and public data. Cora, Citeseer, and PubMed [49] are commonly used public datasets for link prediction. Bold indicates the optimal result in all methods.

Datasets	#Nodes	#Edge	#Features
Cora	2708	8976	1433
Citeseer	3327	7740	303
PubMed	19,717	37,676	500
**Circuits (Ours)**	**34,260**	**55,998**	**32**

**Table 3 sensors-24-00227-t003:** Comparison of experimental results for object detection of custom datasets. Bold indicates the optimal result in all methods.

Algorithm	mAP/%	${\mathrm{mAP}}_{50}$/%	${\mathrm{mAP}}_{75}$/%	AP of Different Categories/%			Model Size
				S	M	L	
YoloX-Tiny	78.8	94.7	88.7	86.8	77.8	86.0	72.8 MB
DDOD	87.3	99.0	96.5	88.4	87.6	89.5	246 MB
Sparse	84.6	97.4	93.8	85.7	84.3	88.0	283 MB
Swin-T	86.0	98.1	94.5	85.5	85.8	88.0	288 MB
**Ours**	**90.3**	**99.0**	**99.7**	**90.3**	**89.8**	**92.9**	328 MB

**Table 4 sensors-24-00227-t004:** Result of graph convolution link prediction. Bold indicates the optimal result in all methods.

Algorithm	Circuits Dataset		AUC of Public Dataset		
	AUC/%	AP/%	Cora	Citeseer	PubMed
DW	80.5 ± 0.01	85.0 ± 0.01	83.1 ± 0.01	80.5 ± 0.02	84.4 ± 0.00
GAE	89.6 ± 0.04	89.9 ± 0.05	91.0 ± 0.02	89.5 ± 0.04	96.4 ± 0.00
VGAE	90.8 ± 0.03	92.0 ± 0.03	91.4 ± 0.01	90.8 ± 0.02	94.4 ± 0.02
ARGE	92.2 ± 0.004	92.8 ± 0.005	92.4 ± 0.003	91.9 ± 0.003	**96.8 ± 0.001**
ARVGE	92.5 ± 0.005	92.6 ± 0.005	92.4 ± 0.004	92.4 ± 0.003	96.5 ± 0.001
RWR-GAE	92.5 ± 0.1	92.8 ± 0.09	92.9 ± 0.3	92.1 ± 0.2	96.2 ± 0.1
RWR-VGAE	92.3 ± 0.2	92.7 ± 0.1	92.6 ± 0.5	92.3 ± 0.3	95.8 ± 0.1
**Ours**	**93.4 ± 0.04**	**93.9 ± 0.05**	**94.8 ± 0.05**	**94.5 ± 0.04**	95.8 ± 0.01

**Table 5 sensors-24-00227-t005:** Ablation experiments of graph convolution. Bold indicates the optimal result in all methods.

Serial No.	Baseline + Improvement			AUC of Different Datasets			
	GAT	SGL	S-Mish	Circuits	Cora	Citeseer	PubMed
Baseline	∘	∘	∘	92.3 ± 0.2	92.6 ± 0.5	92.3 ± 0.3	95.3 ± 0.1
1	*√*	∘	∘	92.9 ± 0.1	93.5 ± 0.2	93.6 ± 0.2	95.6 ± 0.09
2	*√*	*√*	∘	92.9 ± 0.05	94.7 ± 0.04	93.8 ± 0.06	95.7 ± 0.04
3	*√*	*√*	*√*	**93.4 ± 0.03**	**94.8 ± 0.04**	**94.9 ± 0.04**	**95.9 ± 0.03**

## Data Availability

The datasets used or analyzed during the current study are available on 30 December 2023 from the corresponding author on reasonable request.

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
