# Peer review of "Parsing Netlists of Integrated Circuits from Images via Graph Attention Network"

_sensors, 2023, doi:10.3390/s24010227_

Round 1
Reviewer 1 Report
Comments and Suggestions for Authors
Automatically extracting netlists from images containing large numbers of analog IC schematics is a challenging task. It can greatly simplify the design of integrated circuits. In this paper, a bottom-up graph encoder model was presented for automatically parsing schematics in images to obtain the netlist of analog integrated circuits. The lowest composition of the model is an enhanced electronic component detection network based on Swin-Transformer. Experiments were conducted on two custom IC datasets, comprising electronic component detection and component connectivity relationship datasets with 1200 and 3552 samples, respectively. The experimental results show that our proposed algorithm outperforms comparative algorithms on both custom and public datasets. The authors should introduce and cite some recent work on networks and algorithms, such as the titled "Analyses of some structural properties on a class of hierarchical scale-free networks. Fractals, 30(7), 2250136. " and "Single-valued Neutrosophic Eutrosophic Set With Quaternion Information: A Promising Approch to Assess Image Quality." Lastly, the content of the manuscript Conclusion is too limited. The author needs to explain the role and significance of this research.
Comments on the Quality of English Language
Automatically extracting netlists from images containing large numbers of analog IC schematics is a challenging task. It can greatly simplify the design of integrated circuits. In this paper, a bottom-up graph encoder model was presented for automatically parsing schematics in images to obtain the netlist of analog integrated circuits. The lowest composition of the model is an enhanced electronic component detection network based on Swin-Transformer. Experiments were conducted on two custom IC datasets, comprising electronic component detection and component connectivity relationship datasets with 1200 and 3552 samples, respectively. The experimental results show that our proposed algorithm outperforms comparative algorithms on both custom and public datasets. The authors should introduce and cite some recent work on networks and algorithms, such as the titled "Analyses of some structural properties on a class of hierarchical scale-free networks. Fractals, 30(7), 2250136. " and "Single-valued Neutrosophic Eutrosophic Set With Quaternion Information: A Promising Approch to Assess Image Quality." Lastly, the content of the manuscript Conclusion is too limited. The author needs to explain the role and significance of this research.
Reviewer 2 Report
Comments and Suggestions for Authors
For a better understanding of the material, I would suggest the Authors would address the following MINOR points:
A. Fig. 5: What is the impact of the image's resolution (pixel x pixel) on the accuracy of the developed procedure? The Authors are invited to add a few words to the text.
B. B. Fig. 10: At this reviewer, it needs to be clarified the mapping of the upper figures into the lower Feature Tensor. The Authors are invited to add a few words to the text.
Author Response
Thank you for your understanding. Please see the attachment.

Reviewer 3 Report
Comments and Suggestions for Authors
The article presents a new algorithm, a bottom-up graph network encoder model, for extracting a circuit's netlist from an image. The algorithm consists of three stages.
The first stage of the process involves using a Swin-Transformer to detect the electronic components in the circuit. Then, the ports of these components are located, and the results of this step are saved as a graph-based dataset. The last stage involves training a Side Graph Attention Linear Variational Graph Autoencoder (SGL-VGAE) network using the data obtained in the previous stage. This enables the network to predict the connectivity relationship between the components accurately.
Due to the insufficient number of available IC datasets, the authors created a custom IC dataset consisting of electronic component detection and component connectivity relationship data. This custom dataset was then used to evaluate the performance of the algorithm.
I have a few observations regarding the way the information is presented:
- It is a common practice to use acronyms in documents or sections. However, when an acronym is used for the first time, it is important to write the full term first, and then enclose the acronym in parentheses. Both the full term and acronym should be written in bold. In some cases, this rule may be overlooked in the text, and it should be checked and corrected if necessary to ensure consistency.
- On line 107, Kipf's Variational Graph Encoder is referenced without proper bibliographic citation.
- Figure 3 contains six subfigures, but one is missing or incorrectly labeled.
- The bibliographic reference for the Swin-T algorithm is missing from Table 3.
and an observation regarding de experiment results:
- the paper discusses the benefits of one-stage electronic component detection methods, which are known for their speed and real-time applicability. On the other hand, two-stage methods are known for their accuracy. However, Table 3, which compares different algorithms, seems to contain both types of methods. Wouldn't it have been more relevant to compare the detection speed of these methods in addition to their accuracy?
Comments on the Quality of English Language
The language used to present the information is clear and concise.
Author Response

(The authors gave the same response as above.)

Reviewer 4 Report
Comments and Suggestions for Authors
The paper presents a new approach for automatically converting the circuit diagrams into digital netlists. Even though the paper does not provide a breakthrough in image recognition, it proposes several interesting application-specific solutions and analyzes their performance. The paper is well-written and easy to follow. However, several points should be improved.
1. One of the main advantages of the proposed method for the IC design application area is that it can automatically generate a dataset that can be further used to train CNN-based solutions for intelligent component placement. I suggest directly stating this in the abstract because, in another way, the usefulness of the solution for the application area used in the title (IC design) is not clear till the reader reaches the end of the introduction (that may not happen, because the one may decide not to read the full text after the abstract).
2. The authors narrow the application area of their research only to analog IC design, but from my point of view, it is much broader. First, it can generate datasets for AI-assisted PCB design tools like Flux.ai. Second, it will be very useful for retrofitting. Many companies worldwide have expensive equipment accompanied by paper documentation. They want to repair or upgrade these equipment. Still, before doing that, they must put great effort into converting the equipment's documentation into digital. Thus, the paper will only win and gain more readers' interest if the authors cover other possible application areas in the abstract and introduction. In this case, I also recommend avoiding the narrow term "integrated circuits" in the title, as their solution can be used more widely.
3. One of the main issues is whether the paper matches the scope of the MDPI Sensors journal. If we look at the journal's website (https://www.mdpi.com/journal/sensors/about), we will see that image processing/recognition algorithms can fit into scope only when they are used to process data from some sensors. On the contrary, this paper proposes an algorithm that works with image files extracted from PDF, scanned images, and photos. With some imagination, we can assume that the scanner is a sensor, but in this, the influence of the parameters of this sensor should be investigated to make the paper match the journal's scope. Also, it is worth mentioning that the word "sensor" is never used in a paper. In my opinion, it is better to submit this paper to some journal focused on pattern recognition. Still, the authors can also make some changes in the abstract and introduction to better match the scope of the MDPI Sensors journal.
4. The Related Work section is very concise and describes only neural network approaches. At the same time, the first papers on electric circuit recognition and topology analysis were published in the 1980s [a], far before neural networks became a hot topic. In 1993, researchers from Massey University converted scanned electronic circuit schematics automatically to a netlist of the components and their connections [b]. In 1997, a more complex recognition system was proposed by a group of researchers from Lucent Technologies and the University of Nebraska-Lincoln [c]. Since then, multiple approaches have been proposed to perform this task without a neural networks. One of the latest was published in 2015 [d]. After the neural network era started, multiple research papers were presented on hand-written electrical schematic diagram recognition [d-g]. None of these papers were cited in the current research or used as a reference for results comparison, even though they solve much more similar tasks than some papers cited in the Related Work section. The recent papers on electrical schematic diagram recognition (not limited to the ones cited in this comment) should be added to the Related Works and compared with the solutions proposed in the paper.
[a] Okazaki, Akio, et al. "An automatic circuit diagram reader with loop-structure-based symbol recognition." IEEE Transactions on Pattern Analysis and Machine Intelligence 10.3 (1988): 331-341.
[b] Cheng, T., et al. "A symbol recognition system." Proceedings of 2nd International Conference on Document Analysis and Recognition (ICDAR'93). IEEE, 1993.
[c] Yu, Yuhong, Ashok Samal, and Sharad C. Seth. "A system for recognizing a large class of engineering drawings." IEEE Transactions on Pattern Analysis and Machine Intelligence 19.8 (1997): 868-890.
[d] De, Paramita, et al. "Detection of electrical circuit elements from documents images." Document Recognition and Retrieval XXII. Vol. 9402. SPIE, 2015.
[e] Wang, Haiyan, Tianhong Pan, and Mian Khuram Ahsan. "Hand-drawn electronic component recognition using deep learning algorithm." International Journal of Computer Applications in Technology 62.1 (2020): 13-19.
[f] Dey, Mrityunjoy, et al. "A two-stage CNN-based hand-drawn electrical and electronic circuit component recognition system." Neural Computing and Applications 33 (2021): 13367-13390.
[g] Uzair, Waqas, Douglas Chai, and Alexander Rassau. "ElectroNet: An Enhanced Model for Small-Scale Object Detection in Electrical Schematic Diagrams." (2023). (https://assets.researchsquare.com/files/rs-3137489/v1_covered_d1733224-f207-4884-80f4-d65cb9d758aa.pdf?c=1689101568)
5. In section 4.2 and further in section 5.2, authors use bibliographic datasets (Cora, Citescore, PubMed) to analyze the performance of the link prediction of their solution. These datasets are briefly presented in line 276 and Table 2. The inexperienced readers may be misled that these datasets also contain data related to electric circuits. I understand that finding proper datasets for a specific application is difficult. Using datasets from other research areas can be applicable to verify some parts of the complex algorithm (this is what was done in the paper). Still, in this case, authors should provide a detailed description of why these datasets can be used to validate their algorithm. Also, if we analyze Table 2, we see that the chosen bibliographic datasets are quite different from the application-specific dataset created by the authors. They include much fewer nodes and edges but much more features. This difference should also be discussed in section 4.2 in terms of how it can bias the results of further analysis. From my point of view, there is no need to use the bibliographic datasets in the paper because Table 4 already compares the known link prediction algorithms using the application-specific dataset provided by the authors.
6. The references in Table 3 are misleading as they can lead to the interpretation of the results in the row as the one taken from the corresponding papers (and the referenced papers used totally different datasets). I have compared the results presented in papers [39-41] and found them different from those in Table 3. From this and the brief description provided in section 5.1.2, I came to the idea that authors re-implemented neural networks from the papers [39-41], trained them on their dataset, and then compared the results with their solution. If so, authors should directly describe this in a separate paragraph and provide references to the papers in this description. At the same time, these references should be removed from Table 3.
7. It will also be interesting to see the difference between the outputs of the author's solution and the Disentangle your dense object detector (DDOD) [40] in the same way it is provided between Swin+FPN and Swin+DC-PAFPN in Figure 3. DDOD has a different structure compared to Swin+DC-PAFPN, and at the same time, if we analyze Table 3, it provides the second-best result after the method provided in the paper.
8. On line 269, the authors state that the components in the dataset are currently considered to have four ports. If a port is not detected, its value is changed to 0. Thus, this statement can be understood as each component does not have more than four ports. Additional clarifications are needed in this part. Is four ports the limitation of the dataset? Can the electronic component detection algorithm output more than four ports? Why was this limit chosen (ex., operational amplifier by default has five ports)? How difficult would it be to overcome this limitation in the future?
9. The reference [2] in line 15 is unrelated to the paper's topic. It just describes CNN, a well-known term nowadays. I suggest removing it.
Author Response

(The authors gave the same response as above.)

Reviewer 5 Report
Comments and Suggestions for Authors
This article proposes a bottom-up graphic encoder model for automatically parsing schematic diagrams in images, thereby obtaining a netlist of analog integrated circuits. The minimum component of this model is an enhanced electronic component detection network based on Swin Transformer. A component port localization algorithm has been designed, which involves further processing of object detection data and encoding the results into a graph based dataset. Finally, these data are used to train the SGL-VGAE encoding network to achieve prediction of the connection relationships between components. The experiment shows that the algorithm proposed in this article outperforms comparative algorithms on both custom and public datasets. But there are also some issues:
1. The "introduction" section of this article elaborates on a clear and organized structure, with smooth transitions from top to bottom. However, in the introduction section, it is recommended that the author condense it into three key points, highlighting innovation. (Please refer to and cite "Spatial, Spectral, and Texture Aware Attention Network" for details)
The fourth black background image from left to right in Fig1 is not clear. Please provide a clear image and check other images in the article.
3. Regarding the relevant work section, there is a significant gap in the connection between the previous and subsequent content. It is recommended that the author readjust the writing. (Please refer to and reference the "Dual brand collaborative learning network" for details)
The author should provide a more detailed description of the flowcharts for Fig4, Fig6, and Fig7. (Please refer to and reference "ASIF Net: Attention Steered Interweave Fusion Network" for details)
5. The ablation experiment lacks comparison with the latest methods. (Please refer to and reference "CVANet: Cascaded visual attention network" for details)
Comments on the Quality of English LanguageThis article proposes a bottom-up graphic encoder model for automatically parsing schematic diagrams in images, thereby obtaining a netlist of analog integrated circuits. The minimum component of this model is an enhanced electronic component detection network based on Swin Transformer. A component port localization algorithm has been designed, which involves further processing of object detection data and encoding the results into a graph based dataset. Finally, these data are used to train the SGL-VGAE encoding network to achieve prediction of the connection relationships between components. The experiment shows that the algorithm proposed in this article outperforms comparative algorithms on both custom and public datasets. But there are also some issues:
1. The "introduction" section of this article elaborates on a clear and organized structure, with smooth transitions from top to bottom. However, in the introduction section, it is recommended that the author condense it into three key points, highlighting innovation. (Please refer to and cite "Spatial, Spectral, and Texture Aware Attention Network" for details)
The fourth black background image from left to right in Fig1 is not clear. Please provide a clear image and check other images in the article.
3. Regarding the relevant work section, there is a significant gap in the connection between the previous and subsequent content. It is recommended that the author readjust the writing. (Please refer to and reference the "Dual brand collaborative learning network" for details)
The author should provide a more detailed description of the flowcharts for Fig4, Fig6, and Fig7. (Please refer to and reference "ASIF Net: Attention Steered Interweave Fusion Network" for details)
5. The ablation experiment lacks comparison with the latest methods. (Please refer to and reference "CVANet: Cascaded visual attention network" for details)
Author Response

(The authors gave the same response as above.)

Round 2
Reviewer 1 Report
Comments and Suggestions for Authors
The manuscript has been sufficiently improved to warrant publication in Sensors.
Comments on the Quality of English LanguageThe manuscript has been sufficiently improved to warrant publication in Sensors.
Author Response
Please see the attachment. Thank you for your understanding.

Reviewer 4 Report
Comments and Suggestions for Authors
I want to thank the authors for their work. They managed to answer most of my concerns and improved the paper in a very short time. Still, I am not fully satisfied with how the high performance of the DDOD is analyzed in the paper. Indeed, the authors added an additional column to Table 3, but it is not discussed in the paper in any way. In reply to my comments, they briefly mentioned that the small size of DDOD "slightly explains DDOD's high recognition accuracy." However, the reader will read the paper, not the reviews (even when they are open). Previously, I suggested that authors add a Figure similar to Figure 13, which would make it possible to see the difference in the output of the DDOD and the proposed method in the same way it is done for Swin-T. I do not fully insist on this. However, the authors should at least add a separate paragraph comparing the results of DDOD and their approach. I will repeat that this should be done in the paper rather than in the answer to the reviewer's comments. In my opinion, it is incorrect to compare their results with the third-best (Swin-T), ignoring the second-best one (DDOD).
Also, in lines 19, 74, and 347, there are '-' symbols at the beginning of the lines, which are highly like typos. I suggest removing them.
Author Response

(The authors gave the same response as above.)
